# Nonspecific Inhibition of IL6 Family Cytokine Signalling by Soluble gp130

**DOI:** 10.3390/ijms25031363

**Published:** 2024-01-23

**Authors:** Anissa A. Widjaja, Stuart A. Cook

**Affiliations:** 1Cardiovascular and Metabolic Disorders Program, Duke-National University of Singapore Medical School, 8 College Road, Singapore 169857, Singapore; 2National Heart Centre Singapore, National Heart Research Institute Singapore, Singapore 169609, Singapore; 3MRC-London Institute of Medical Sciences, Hammersmith Hospital Campus, London W6 8RF, UK

**Keywords:** IL6, gp130, IL11, trans-signaling, cis-signaling, STAT3, OSM, Olamkicept, sgp130Fc

## Abstract

IL6 is a proinflammatory cytokine that binds to membrane-bound IL6 receptor (IL6R) or soluble IL6R to signal via gp130 in *cis* or *trans*, respectively. We tested the hypothesis that sgp130Fc, which is believed to be a selective IL6 *trans*-signalling inhibitor, is in fact a non-specific inhibitor of gp130 signalling. In human cancer and primary cells, sgp130Fc inhibited IL6, IL11, OSM and CT1 *cis*-signalling. The IC_50_ values of sgp130Fc for IL6 and OSM *cis*-signalling were markedly (20- to 200-fold) lower than the concentrations of sgp130Fc used in mouse studies and clinical trials. sgp130 inhibited IL6 and OSM signalling in the presence of an ADAM10/17 inhibitor and the absence of soluble IL6R or OSMR, with effects that were indistinguishable from those of a gp130 neutralising antibody. These data show that sgp130Fc does not exclusively block IL6 *trans*-signalling and reveal instead that broad inhibition of gp130 signalling likely underlies its therapeutic effects. This proposes global or modular inhibition of gp130 as a therapeutic approach for treating human disease.

## 1. Introduction

Interleukin 6 (IL6) is a proinflammatory cytokine that binds membrane-bound IL6R to signal in *cis* via the shared glycoprotein 130 (gp130) receptor. Soluble IL6 receptor (sIL6R), which is released from cell membranes by ADAM proteases, can also bind to IL6 to form IL6:sIL6R complexes that amplify IL6 signalling in autocrine and activate gp130 in cells lacking IL6R, in paracrine [1,2,3,4]. This secondary signalling mechanism is termed *trans*-signalling and is thought to be strongly pro-inflammatory and disease causing, as opposed to IL6 *cis*-signalling that is believed homeostatic [2]. gp130 itself can be cleaved from the plasma membrane to form soluble gp130 (sgp130) that acts as an inhibitor of IL6 *trans*-signalling by decoying/trapping IL6:sIL6R complexes [5].

The field of IL6 *trans*-signalling relies heavily on experiments using a synthetic protein i.e., sgp130Fc [6] that comprises two sgp130 molecules linked to an IgG1 antibody Fc domain. sgp130Fc is taught as a specific inhibitor of IL6 *trans*-signalling that leaves IL6 *cis*-signalling intact. sgp130 administered to rodents is protective against diseases in multiple mouse and rat models, with superiority over anti-IL6/IL6R monotherapy in some instances [7,8,9]. Following proof-of-concept (POC) therapeutic studies in rodents, sgp130Fc was developed further as an anti-inflammatory drug (olamkicept), which is now in clinical trials [2,6,10,11]. sgp130Fc has been described as “the first selective inhibitor of IL6 trans-signalling” [2] and “a decoy protein that exclusively blocks IL6 proinflammatory trans-signalling” [11].

While sgp130 is widely recognized and accepted as a specific inhibitor of IL6 *trans*-signalling, it was suggested to inhibit Oncostatin M (OSM) and Leukaemia Inhibitory Factor (LIF) signalling when used at high concentrations in early publications [1,5,6,12]. We postulated that sgp130Fc might—at high concentration—act as a decoy for membrane bound IL6:IL6R and/or Interleukin 11 (IL11):IL11RA *cis*-signalling complexes. Additionally, sgp130 may also directly serve as a decoy for OSM, which is unable to signal in *trans* as it does not bind to its alpha receptors until after it has formed a low affinity complex with gp130 [13,14,15,16].

Here, we tested the hypothesis that sgp130Fc, at concentrations relevant to those used in mouse experiments and in clinical trials [7,10,11,17], is a non-specific inhibitor of IL6 family cytokine signalling, rather than being a selective inhibitor of IL6 *trans*-signalling.

## 2. Results

### 2.1. sgp130Fc Inhibits IL6, IL11, OSM, and CT1 Signalling

We tested our hypothesis by conducting experiments in a lung epithelial cancer cell line (A549), primary human hepatocytes and primary human hepatic stellate cells (HSCs, a stromal cell type), which express the diversity of IL6 family cytokine receptors, including IL11RA. This approach helps to avoid potential bias in IL6 family receptor expression that might occur in a single cell type and tests whether detected effects are cell-type specific or more generalisable.

For IL6 or IL11 to signal in *trans*, IL6R or IL11RA must first be cleaved from the plasma membrane, which is primarily achieved by ADAM10/17 [3,4]. We screened the supernatants of cells incubated (24 h) in media for the presence of sIL6R and sIL11RA and extended analyses to sOSMRβ (sOSMR) and sLIFR (Figure 1A). sIL6R was only detected in A549 supernatants with levels below the lower limit of detection (LLOD: 15.2 pg/mL) in hepatocytes and HSCs. sIL11RA was abundant in all supernatants with highest levels seen for hepatocytes, sOSMR was undetectable in all supernatants (LLOD: 0.094 ng/mL), and sLIFR was equally detected in A549 and hepatocyte conditioned media but not in HSC media (LLOD: 0.188 ng/mL) (Figure 1A).

In the presence or absence of sgp130Fc (5 µg/mL), we assessed the phosphorylation of STAT3 (*p*-STAT3) in response to IL6 family cytokines that have been reported to signal in *trans* (IL6 and IL11), cytokines not known to signal in *trans* (Cardiotrophin 1 (CT1), ciliary neurotrophic factor (CNTF), and LIF), and an IL6 family cytokine previously reported as incapable of *trans*-signalling (OSM).

While there was some variation in the *p*-STAT3/STAT3 response to cytokine stimulation, results were similar across cell types (Figure 1B–D and Appendix A). For IL6, IL11, CT1 and OSM, *p*-STAT3/STAT3 ratios were significantly inhibited by sgp130Fc. CNTF and LIF-stimulated STAT3 activation were unaffected by the presence of sgp130Fc.

It was notable that following OSM stimulation (15 min), total STAT3 levels were diminished and *p*-STAT3/STAT3 ratios notably increased, which would be consistent with phosphorylation-induced protein degradation secondary to strong phosphorylation of STAT3 by OSM, as compared to other IL6 family cytokines [18,19]. The specificity of this effect was confirmed by probing samples stimulated with OSM for GAPDH, which was unchanged.

These data showed that sgp130Fc at a concentration of 5 µg/mL inhibits gp130-mediated STAT3 activation by varied IL6 family cytokines, including OSM that does not signal in *trans* [13,14,15,16]. Furthermore, effects of sgp130Fc on STAT3 activation were unrelated to the presence, abundance, or absence of soluble receptors for IL6 family cytokines, suggesting *trans* effects are not a major component of the signalling we observed.

### 2.2. sgp130FC Inhibits IL6 and OSM Cis-Signalling

To further determine if sgp130Fc inhibits IL6 *cis*-signalling we pre-incubated A549 cells with either vehicle control (0.1% DMSO) or GW280264X (1 μM in 0.1% DMSO, 16 h), which potently inhibits ADAM10 and 17 (IC_50_, 8.0 nM and 11.5 nM, respectively), to prevent IL6R cleavage and sIL6R generation. On the day of the experiment, cells were washed with PBS, supplemented with fresh media, again with GW280264X present or not, and immediately stimulated (15 min) cells with IL6 with or without the addition of sgp130Fc (5 µg/mL).

Immunoblotting for *p*-STAT3/STAT3 revealed robust STAT3 activation in cells incubated with or without GW280264X and equally potent inhibition of this effect by sgp130Fc. At the end of the experiment, supernatants were assessed for sIL6R that was detectable in the absence of GW280264X but not detected in media from cells incubated with GW280264X. (Figure 2A,B and Appendix A). These data reinforce the fact that sgp130Fc inhibits IL6 *cis*-signalling.

Experiments using GW280264X were repeated in primary human hepatocytes to exclude the possibility of an undetected presence [and effect] of IL6R in hepatocyte media, despite the negative ELISA results (Figure 1A). Comparable levels of STAT3 activation were observed following IL6 stimulation of DMSO (control) and GW280264X-treated hepatocytes, which reconfirmed the absence of an IL6:sIL6R *trans*-signalling effect in hepatocyte media (Figure 2A,B). There was similar inhibition of STAT3 activation by sgp130Fc in the presence or absence of ADAM10/17 inhibitor, excluding a *trans*-signalling component (Figure 2A,B).

We then explored the effects of sgp130Fc on IL6 and OSM *cis*-signalling in A549 cells across a dose range of 1.25–100 µg/mL (Figure 2C–F). Given the presence of IL6R in A549 media (Figure 1A), we conducted experiments in this cell line in the presence of GW280264X to obviate possible IL6 *trans*-signalling effects. sgp130Fc dose-dependently reduced IL6 and OSM-mediated *cis*-stimulation of STAT3 activation. The relationship between sgp130Fc concentrations and the levels of *p*-STAT3 was evaluated using semi-quantitative densitometry, which revealed IC_50_ values of 7.73 μg/mL and 1.06 μg/mL for sgp130Fc inhibition of IL6 and OSM *cis*-signalling, respectively (Figure 2D,F).

To address any potential bias arising from the effects of sgp130Fc in a single cell type, particularly a cancer cell line such as A549, we evaluated the IC_50_ values of sgp130Fc for IL6 *cis*-signalling in primary human hepatocytes and HSCs. Consistent with the results in A549 cells, a dose-dependent reduction in IL6-mediated phosphorylation of STAT3 was observed with sgp130Fc in both cell types with IC_50_ values of 3.83 μg/mL and 5.21 μg/mL in hepatocytes and HSCs, respectively (Figure 2G,H).

As mentioned in the introduction, sgp130Fc has pronounced therapeutic effects across murine disease models and also shows promise in human trials. On review of the literature, we observed that many of the murine POC studies used sgp130Fc at concentrations in excess of its IC_50_ values for IL6 or OSM *cis*-signalling, which was also true for clinical trials of olamkicept (Table 1). This suggests that sgp130Fc, at concentrations used in preclinical studies and clinical trials, most likely inhibits gp130 signalling beyond just IL6 *trans*-signalling.

We next examined the effects of sgp130Fc, as compared to a commercially available gp130 neutralising antibody (clone B-R3) [27]). In A549 cells treated with GW280264X, B-R3 inhibited IL6 and OSM signalling, as expected. sgp130Fc also inhibited both IL6 and OSM signalling with effects that were indistinguishable from those of B-R3 (Figure 3A–C).

To conclude, we present a simplified overview and interpretation of the findings of this study in a schematic, which illustrates the inhibitory effects of sgp130Fc on IL6 *trans*-signalling at low concentrations, and on IL6 family cytokine *cis-*signalling at high concentrations (Figure 3D).

## 3. Discussion

It is taught that IL6 *cis*-signalling is homeostatic, whereas IL6 *trans*-signalling amplifies inflammatory responses in autocrine and paracrine and is disease-causing [2]. As such, sgp130Fc is proposed, and has been shown, to have therapeutic benefits over combined IL6 *cis-* and *trans*-signalling inhibitors, such as tocilizumab [7]. However, it has also been reported that sgp130 might inhibit some aspects of OSM, CNTF and/or LIF signalling [1,5,6,12] and IL11 *trans*-signalling [28].

We show here that sgp130Fc inhibits gp130-dependent *cis*-signalling by several IL6 family cytokines in neoplastic (A549), stromal (hepatic stellate) and epithelial (hepatocyte) cells. This broad inhibitory effect across diverse cell types is independent of the presence, absence, or amount of sIL6R or sIL11RA, for IL6 or IL11 signalling, and was also seen for CT1 and OSM. While proponents of IL6 *trans*-signalling may argue that sub-picomolar levels of IL6R may still play a role, our findings also reveal that the sgp130Fc activity is not influenced by the presence of GW280264X, an inhibitor for IL6R shedding. The inhibitory effects of sgp130Fc were particularly robust for OSM, which does not signal in *trans*. This may be attributed to OSM’s direct and low-affinity binding to gp130 before interacting with its alpha receptors, potentially rendering it more susceptible to decoying [13,14,15,16].

The investigational drug olamkicept shows suggestive therapeutic effect (NCT03235752) at a dose of 600 mg, bi-weekly: maximum concentration (C_max_), 159 μg/mL [10,11]. These C_max_ levels are approximately 100-fold higher than those reported as specific for IL6 *trans*-signalling [6] and, assuming equitable drug distribution in the extracellular space, are expected to inhibit IL6, IL11, OSM and CT1 *cis*-signalling. Similarly, the concentrations of sgp130Fc that have therapeutic effects in murine POC studies exceed the IC_50_ values of sgp130Fc for IL6 family cytokine *cis*-signalling.

Our data suggest the following: [1] sgp130Fc, at doses effective in treating both murine and human disease, is not specific for IL6 *trans*-signalling and [2] sgp130Fc effects on IL6 family cytokine *cis*-signalling are similar to those of a gp130 neutralising antibody. These findings raise important questions for the field of IL6 *trans*-signalling in general that are amplified by recent data indicating the non-specific nature of HyperIL6, a synthetic trans-signalling complex [29].

We thus propose global inhibition of gp130 *cis*-signalling (e.g., using bazedoxifene [30] or an antibody like B-R3 [27]) as a therapeutic approach for treating diseases in which sgp130Fc, an incomplete blocker of gp130 *cis*- and *trans*-signalling, has demonstrated efficacy in preclinical and clinical studies. The mouse model of cytokine storm syndrome (CSS) or sepsis may be particularly useful for testing this hypothesis, given the important roles of IL6 [17,31] and OSM [32] in CSS pathogenesis. Notably, genetic deletion of gp130 or administration of sgp130Fc has shown protective effects against CSS [17,33,34], gp130 is upregulated in humans and mice with CSS [35], and genetically enhanced gp130 signalling causes hypersensitivity and increased mortality in CSS [36].

Genetic perturbation of gp130 signalling during development and early life has well-documented detrimental effects in mice and humans [37,38]. Based on this observation, gp130 has been predominantly viewed as a protective factor. However, more recent studies suggest that inhibition of gp130 in adult mice can protect against various conditions, including hepatotoxicity, sepsis, osteoarthritis, inflammation, myeloma, and endothelial dysfunction, while also promoting tissue regeneration [9,34,39,40].

In light of our new findings, global or modular inhibition of gp130 signalling in adults could be considered as a therapeutic approach for human diseases where enhanced gp130 activity is specifically implicated. Such conditions include inflammatory bowel disease [41], solid cancers [42], myeloma [9], retinal disease [8], and autoimmune diseases [43]. It is important to consider potential on-target toxicities that may be associated with prolonged inhibition of gp130, such as infection risk, for long-term indications. Interestingly, sgp130Fc has a reassuring safety profile, including the absence of platelet toxicity, even when used at concentrations expected to inhibit OSM signalling, which is intriguing considering that direct targeting of OSM causes thrombocytopenia [44].

## 4. Materials and Methods

### 4.1. Antibodies

For neutralisation study: anti-gp130 (B-R3, ab34315, Abcam, Cambridge, UK), IgG control (clone 11E10, Genovac, Freiburg, Germany).

For immunoblotting: GAPDH (2118, CST, Danvers, MA, USA), phospho-STAT3 (4113, CST, Danvers, MA, USA), STAT3 (4904, CST, Danvers, MA, USA), mouse HRP (7076, CST, Danvers, MA, USA), rabbit HRP (7074, CST, Danvers, MA, USA). All primary antibodies were used at 1:1000 dilution in TBST, and secondary antibodies were diluted 1:2000 in TBST containing 3% bovine serum albumin (BSA).

### 4.2. Recombinant Proteins

Recombinant human (rh)CNTF (257-NT, R&D Systems, Minneapolis, MN, USA), rhCT1 (612-CD, R&D Systems, Minneapolis, MN, USA), rhIL6 (206-IL, R&D Systems, Minneapolis, MN, USA), rhIL11 (Z03108, Genscript, Minneapolis, MN, USA), rhLIF (7734-LF, R&D Systems, Minneapolis, MN, USA), rhOSM (PHC5015, ThermoFisher Scientific, San Francisco, CA, USA), sgp130Fc (671-GP, R&D Systems, Minneapolis, MN, USA). All lyophilized protein was dissolved in PBS with 15 min of gentle agitation.

### 4.3. Chemicals

Bovine serum albumin (BSA, A7906, Sigma-Aldrich, St. Louis, MO, USA), DMSO (D2650, Sigma-Aldrich, St. Louis, MO, USA), GW280264X (HY-115670, MedChemExpress, South Brunswick, NJ, USA), Poly-L-Lysine solution (P4707, Sigma-Aldrich, St. Louis, MO, USA).

### 4.4. Cell Culture

All cells used in this study were sourced commercially. Cells were grown and maintained at 37 °C and 5% CO_2_. The growth medium was renewed every 2–3 days and cells were passaged at 80% confluence, using standard trypsinization techniques. Cells were serum-starved overnight in basal media prior to stimulation. Cells were stimulated with either recombinant human CNTF, CT1, IL6, IL11, LIF, or OSM, at a concentration of 5 ng/mL in serum-free media for 15 min. Stimulated cells were compared to unstimulated cells that have been grown for the same duration under the same conditions, but without the stimulus (referred to as ‘baseline (BL)’ conditions).

#### 4.4.1. A549

Human lung epithelial carcinoma cells (A549) that were isolated from a 58-year-old male (CCL-185, ATCC) were grown and maintained in DMEM (11995065, ThermoFisher Scientific, San Francisco, CA, USA) complete medium which contains 10% FBS (10500064, ThermoFisher Scientific, San Francisco, CA, USA) and 1% P/S (15140122, ThermoFisher Scientific, San Francisco, CA, USA).

#### 4.4.2. Human Hepatocytes

Primary human hepatocytes used in this study were isolated from a 22-week-old foetus (5200, ScienCell, Carlsbad, CA, USA) and cultured as described previously [45]. Briefly, once hepatocytes were recovered from the initial thaw cycle, cells were seeded at a density 4 × 10^5^ cells/well in hepatocyte medium (5201, ScienCell, Carlsbad, CA, USA) supplemented with 2% fetal bovine serum, 1% penicillin-streptomycin on a a poly-L-lysine-coated 6-well plate (2 µg/cm^2^, 0403, ScienCell, Carlsbad, CA, USA). Hepatocytes were used directly for downstream experiments within 48 h of seeding. All experiments with primary human hepatocytes were carried out at passage 2 (P2).

#### 4.4.3. Human Hepatic Stellate Cells (HSC)

Primary HSC used in this study were isolated from a 50-year-old male (5300, ScienCell, Carlsbad, CA, USA) and cultured as described previously [46]. For experiments, we thawed and seeded P3 cells at a density of 2.5 × 10^5^ cells/well on a poly-L-lysine-coated 6-well plate in stellate cells complete media (5301, ScienCell, Carlsbad, CA, USA). Within 24 h of seeding, HSC were serum-starved overnight prior to stimulation.

### 4.5. ADAM10/17 Inhibitor Experiments

GW280264X came pre-dissolved in 100% DMSO at a concentration of 10 mM, which we subsequently diluted to a stock concentration of 1 mM in 100% DMSO DMSO and kept as stock. Further dilution was performed to a final concentration of 1 µM i.e., 1 µL of GW280264X in 1 mL of cell culture media. Cells were treated with 1 µM of GW280264X throughout the overnight starvation, a 30-min incubation with sgp130, and a 15-min stimulation period; 0.1% DMSO was used as control.

### 4.6. sgp130/Anti-gp130 Experiment

sgp130Fc was dissolved in PBS to a stock concentration of 2.5 mg/mL; IgG and anti gp130 (B-R3) were dissolved in PBS to a stock concentration of 5 mg/mL. Cells were pre-incubated with either IgG (5 µg/mL), sgp130Fc (100 µg/mL) or B-R3 (5 µg/mL) for 30 min prior to cytokine stimulation.

### 4.7. IC_50_ Experiments

Cells were pre-incubated with either sgp130Fc (1.25, 2.5, 5, 10, 20, 40, 80, 100 µg/mL) an equivalent amount of PBS to 100 ug/mL of sgp130Fc (40 µL in 1 mL media) for 30 min and then stimulated with 5 ng/mL of various cytokines (as outlined in the text/figure/figure legends) for 15 min.

### 4.8. Immunoblotting

Western blots were carried out on total protein extracts from A549, hepatocytes, and HSCs. After stimulation, cells were rinsed twice in cold PBS. Cell lysates were homogenised in radioimmunoprecipitation assay (RIPA) Lysis and Extraction Buffer (89901, Thermo Fisher Scientific, San Francisco, CA, USA) containing protease inhibitor (A32965, Thermo Fisher Scientific, San Francisco, CA, USA), and phosphatase inhibitor (A32957, Thermo Fisher Scientific, San Francisco, CA, USA), followed by centrifugation to clear the lysate. Protein concentrations were determined by Bradford assay (5000006, Bio-Rad, Watford, UK). Equal amounts (5 µg) of protein lysates were separated by SDS-PAGE and transferred to PVDF membranes. Protein bands were visualised using the SuperSignal West Femto Maximum Sensitivity Substrate detection system (34096, Thermo Fisher Scientific, San Francisco, CA, USA) with the appropriate secondary antibodies: anti-mouse HRP or anti-rabbit HRP. Densitometry analyses were performed by ImageJ software. (version 1.53f51), in which the phospho STAT3 signal was normalised to its respective total STAT3 intensity. For IC_50_ experiment (Section 4.7), we considered STAT3 activation levels at baseline as maximal inhibition (100%) while STAT3 activation levels after stimulation with the respective recombinant protein (IL6/IL11;10 ng/mL) in the presence of PBS constituted minimum inhibition (0%).

### 4.9. Enzyme-Linked Immunosorbent Assay (ELISA)

The levels of IL6R, IL11RA, LIFR, and OSMR in an equal volume of cell culture supernatant were quantified using Human IL-6R alpha Quantikine ELISA kit (DR600, R&D Systems, Minneapolis, MN, USA), Human interleukin 11 receptor alpha (IL11RA) ELISA Kit (MBS453817, MyBioSource, San Diego, CA, USA), Human LIFR ELISA Kit (MBS2513036, MyBioSource, San Diego, CA, USA), or Human OSMR ELISA Kit (MBS2514091, MyBioSource, San Diego, CA, USA), respectively. All ELISAs were performed according to the manufacturer’s protocol.

### 4.10. Statistical Analysis

All statistical analyses were performed using GraphPad Prism software (version 9). Data were tested for normality with Shapiro-Wilk tests and all were found to follow normal distribution. One-way ANOVA with Tukey’s correction was used when several conditions were compared to each other. The criterion for statistical significance was set at *p* < 0.05.

## 5. Conclusions

In summary, the conventional notion of sgp130 as “*a decoy protein that exclusively blocks IL6 proinflammatory trans-signalling*” [11] appears invalid at the concentrations of sgp130Fc used in most rodent studies and in human clinical trials. Instead, the strong therapeutic effects of sgp130Fc more likely reflect its more extensive inhibition of IL6, IL11, OSM and CT1 *cis*-signalling, which would explain its superiority over anti-IL6 or anti-IL6R monotherapy [7,8,9]. An implication of this interpretation is that compounds, whether small or large molecules, that have global or modular inhibition of gp130, which is superior or more extensive to that achieved with sgp130Fc, could offer a new therapeutic approach for treating human disease.

## Figures and Tables

**Figure 1 ijms-25-01363-f001:**
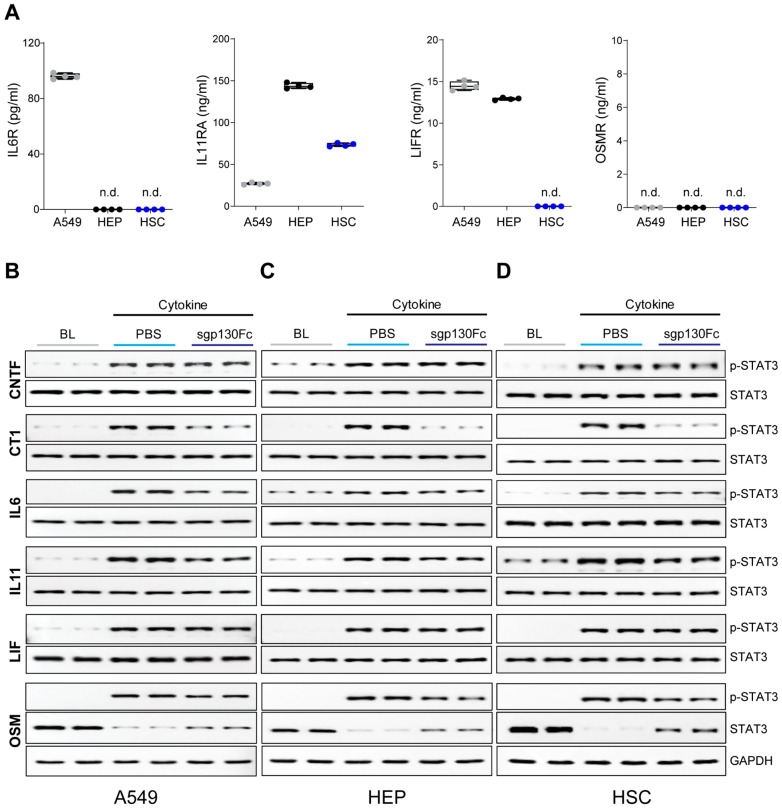
sgp130Fc inhibits IL6, IL11, OSM, and CT1-induced STAT3 activation. (**A**) ELISA of IL6R, IL11RA, LIFR, and OSMR in the conditioned supernatant of A549, primary human hepatocytes (HEP), and primary human hepatic stellate cells (HSC) 17 h after plating. Data are shown as box-and-whisker plots with median (middle line), 25th–75th percentiles (box) and min-max percentiles (whiskers). Western blots of *p*-STAT3 and STAT3 in CNTF, CT1, IL6, IL11, LIF (5 ng/mL)-stimulated (**B**) A549, (**C**) HEP, (**D**) HSC (15 min) and (**B**–**D**) immunoblots of *p*-STAT3, STAT3, GAPDH in OSM (5 ng/mL)-stimulated A549, HEP, HSC (15 min) in the presence or absence of sgp130Fc (5 µg/mL). (**A**–**D**) *n* = 4 biological replicates. BL: baseline. n.d. not detected.

**Figure 2 ijms-25-01363-f002:**
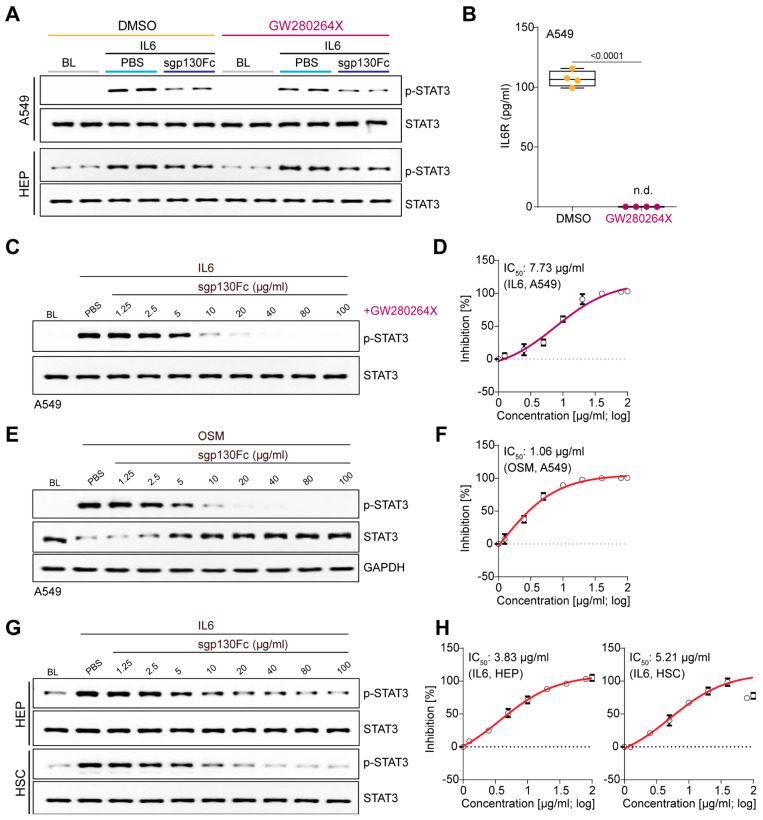
Dose-dependent inhibition of IL6 and OSM *cis*-signalling by sgp130Fc. (**A**) Western blots of STAT3 activation status (*p*-STAT3/STAT3) in A549 and HEP in the following conditions: without stimulus (BL), IL6 + PBS, or IL6 + sgp130Fc in the presence of either DMSO or GW280264X. (**B**) ELISA of IL6R in the supernatant of A549 after 17 h incubation with either DMSO or GW280264X. Data are shown as box-and-whisker plots with median (middle line), 25th–75th percentiles (box) and min-max percentiles (whiskers); 2-tailed Student’s *t*-test. (**C**) Western blots and (**D**) densitometry analyses of *p*-STAT3/STAT3 in GW280264X-treated A549 following stimulation with IL6 in the presence of different concentrations of sgp130Fc. (**E**) Western blots of *p*-STAT3, STAT3, and GAPDH, and (**F**) densitometry analyses of *p*-STAT3/STAT3 in OSM-stimulated A549 in the presence of increasing concentrations of sgp130Fc. (**G**) Western blots and (**H**) densitometry analyses of *p*-STAT3/STAT3 in IL6-stimulated HEP and HSC in the presence of increasing concentrations of sgp130Fc. (**D**,**F**,**H**) Data are shown as mean ± s.d. (**A**–**H**) *n* = 4 biological replicates; DMSO (0.1%), GW280264X (1 µM), IL6 (5 ng/mL), OSM (5 ng/mL), sgp130Fc (5 µg/mL, unless otherwise specified). BL: baseline. n.d. not detected.

**Figure 3 ijms-25-01363-f003:**
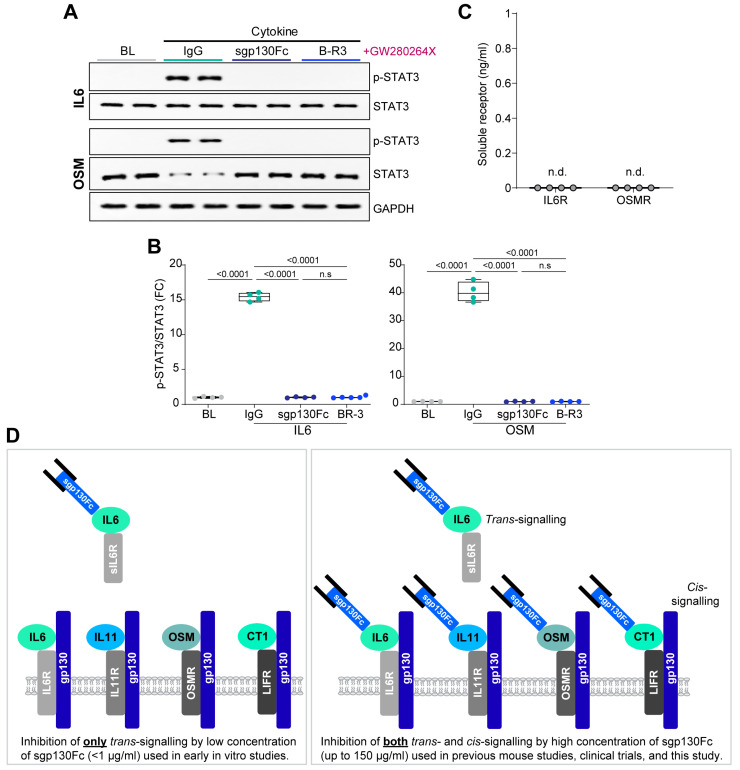
Comparison of the inhibitory effects of sgp130Fc versus a gp130 neutralising antibody on IL6 or OSM *cis*-signalling.(**A**) Western blots of *p*-STAT3, STAT3 (for IL6 and OSM), and GAPDH (for OSM) and (**B**) densitometry analyses of *p*-STAT3/STAT3 in GW280264X-treated A549 following stimulation with IL6, or OSM in the presence of either IgG, sgp130Fc, or a neutralising antibody against gp130 (clone B-R3). (**C**) Levels of IL6R, and OSMR protein in the supernatant from GW280264X-treated A549 as quantified by ELISA. (**D**) Schematic showing our proposed interpretation of the comparative effects between low versus high concentrations of sgp130Fc on the inhibition of gp130-mediated *cis*- and/or *trans-*signalling. (**A**–**C**) *n* = 4 biological replicates; B-R3 (5 µg/mL), GW280264X (1 µM), IgG (5 µg/mL), IL6 (5 ng/mL), IL11 (5 ng/mL), OSM (5 ng/mL), sgp130Fc (100 µg/mL). (**B**,**C**) Data are shown as box-and-whisker plots with median (middle line), 25th–75th percentiles (box) and min-max percentiles (whiskers). (**B**) One-way ANOVA with Tukey’s correction. BL: baseline; FC: fold change. n.d. not detected.

**Table 1 ijms-25-01363-t001:** Examples of pre-clinical and clinical studies where sgp130Fc was given at doses that exceeded its half-maximal inhibitory (IC_50_) concentrations for IL6 or OSM *cis*-signalling. Dose per mouse reports published values or is estimated based on an average adult mouse mass of 25 g. Estimated maximum concentrations for mouse studies were calculated based on the assumption that sgp130Fc is confined and equally distributed across the extracellular fluid spaces (interstitial fluid and plasma), which represents ~20% of the body weight of an adult mouse [20]. i.p.: intraperitoneal; i.v.: intravenous, d: day.

Disease Model or Clinical Trial	Dose, Route, Frequency	Max Dose Per Individual	Estimated Maximum Concentration (μg/mL)	IC_50_ for IL6 or OSM *cis*- Signalling Exceeded	Reference
Murine Studies					
Polymicrobial sepsis	10 mg/kg, i.p., 7 d	250 μg	50	Y	[17]
Colitis	-	500 μg	100	Y	[21]
Arthritis	2.5 mg/kg, i.p., 7 d	62.5 μg	12.5	Y	[22]
Colon cancer	-	500 μg	100	Y	[23]
Nephritis	3 mg/kg, i.p., 3.5 d	75 μg	15	Y	[24]
Myocardial infarction	0.5 mg/kg, i.p.	12.5 μg	2.5	Y	[7]
Atrial Fibrillation	0.5 mg/kg, i.p., 3.5 d	12.5 μg	2.5	Y	[25]
Diabetic retinopathy	5 mg/kg, i.p., 3.5 d	125 μg	25	Y	[26]
Human Studies					
Patients with inflammatory bowel disease	600 mg, i.v., 14 d	600 mg	>100	Y	[11]
Patients with ulcerative colitis	300 mg or 600 mg, i.v., 14 d	600 mg	159	Y	[10]

## Data Availability

All data are provided in the manuscript and Appendix A.

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
