# Peer review of "Nonspecific Inhibition of IL6 Family Cytokine Signalling by Soluble gp130"

_ijms, 2024, doi:10.3390/ijms25031363_

Round 1
Reviewer 1 Report
Comments and Suggestions for Authors
Please explain :
1. why authors choose 3 cell types and not others ( as described on - 2.4.1.,2.4.2., and 2.4.3.) . Are they the most representatives for such kind of cell line based experiment or rather for further preclinical or clinical interpretations/results application? Or both?
Reviewer 2 Report
Comments and Suggestions for Authors
1. All WB images lacked internal control and the total STAT protein was lighter in some treatment groups, which may affect the result interpretation. The authors need to discuss whether the corresponding interventions directly affect the expression level of STAT or other possible factors contributing to the results and add the internal control situation.
2. STAT phosphorylation only reflects the level of IL6 pathway activation. whether the classical or trans signals of IL6 can be directly detected by immunofluorescence and other methods. Currently only STAT phosphorylation level is available as a detection index, which does not provide good support for the underlying mechanism, and authors should consider this point.
3. The authors should explain why A549, HEP, and HSC cells were chosen for the experiment, and analyze the differences between the different cell lines in the discussion.
4. the discussion is not in-depth enough.
Comments on the Quality of English LanguageThe language is fine.
Round 2
Reviewer 2 Report
Comments and Suggestions for Authors 1. In fact, I would like to point out that the study needs to standardize the experimental mapping, providing homogeneous images of internal control proteins (GAPDH or beta-actin) can support the authenticity of the data, or do the authors believe that the results of the internal control are not necessary to show? 2. Based on the systemic considerations of the study, the authors could still have chosen other molecules directly or indirectly downstream of the gp130 subunit signaling for validation as supplementary material to jointly illustrate the level of activation of the pathway, with the above being my reservation. Comments on the Quality of English LanguageEnglish is OK
